# Cortical neurons exhibit diverse myelination patterns that scale between mouse brain regions and regenerate after demyelination

Cody L. Call[1] & Dwight E. Bergles [1,2✉]

Axons in the cerebral cortex show a broad range of myelin coverage. Oligodendrocytes establish this pattern by selecting a cohort of axons for myelination; however, the distribution of myelin on distinct neurons and extent of internode replacement after demyelination remain to be defined. Here we show that myelination patterns of seven distinct neuron subtypes in somatosensory cortex are influenced by both axon diameter and neuronal identity. Preference for myelination of parvalbumin interneurons was preserved between cortical areas with varying myelin density, suggesting that regional differences in myelin abundance arises through local control of oligodendrogenesis. By imaging loss and regeneration of myelin sheaths in vivo we show that myelin distribution on individual axons was altered but overall myelin content on distinct neuron subtypes was restored. Our findings suggest that local changes in myelination are tolerated, allowing regenerated oligodendrocytes to restore myelin content on distinct neurons through opportunistic selection of axons.

[1] The Solomon Snyder Department of Neuroscience, Johns Hopkins University, Baltimore, MD, USA. [2] Johns Hopkins University, Kavli Neuroscience Discovery Institute, Baltimore, MD, USA. ✉email: dbergles@jhmi.edu

Circuits within the cerebral cortex are formed by a diverse population of neurons that integrate input from both local axon collaterals and long-range projections within specific lamina. In the uppermost layer of the cortex (layer I), axonal projections synapse onto the dendritic tufts of pyramidal neurons, serving as a critical hub for integrating local inhibition, thalamocortical and corticocortical excitation, and long-range neuromodulation, providing top-down regulation of sensory inputs[1–3]. Despite the critical need to integrate input from both local and distant inputs in this region, and the ability of myelin to control the timing of synaptic activity, we have only a limited understanding of the myelin patterns along axons of distinct neuron subtypes that course within this cortical area. Although a small fraction of these axons are well-myelinated and extend within layer I for many millimeters in rodents (or centimeters, in primates) before terminating[4,5], the overall myelin content within each cortical region is small[6], suggesting a high level of specificity in oligodendrocyte selection of axons. Approximately 10% of myelin sheaths in layer I surround inhibitory axons, the majority of which express parvalbumin (PV)[7,8], but the vast majority are associated with an unknown population of neurons. Moreover, individual axons in the cortex can vary extensively in the number and pattern of myelin sheaths along their lengths[9–11]. This "discontinuous" myelination has been well-described on PV interneurons and proximal axon segments of pyramidal neurons[8,11], but the pattern of myelin along these and other types of neurons has not been explored in layer I.

Schwann cells in the peripheral nervous system select axons for myelination strictly by axon diameter[12], with a threshold of ~1.5 µm[12–14]. Similarly, oligodendrocytes in the CNS only myelinate axons or inert fibers in vitro with diameters > 0.3 µm[15–17], suggesting that there are strict biophysical constraints on which axons can be wrapped. Nevertheless, individual oligodendrocytes are capable of myelinating a range of axon sizes[18] and provide sheaths of different thicknesses and lengths, often correlated with the diameter of each axon[18,19], illustrating the additional complexity of CNS myelination. The discontinuity of myelin along individual axons and the numerous axons of permissive diameter that lack myelin indicate that oligodendrocytes in gray matter may be influenced by other factors in addition to axon diameter, such as spiking activity[20–24] or surface protein expression[25] to yield cell-type-specific myelination patterns.

The complex patterns of myelination that exist in the cortex create significant challenges for repair. Demyelinating lesions of the gray matter in multiple sclerosis (MS) most commonly occur just below the pia[26–28], and are associated with cognitive disabilities and poorer prognosis[26,29–33], making this region of particular interest for understanding the identities of the axons myelinated and the extent to which they are remyelinated in recovery. Cortical PV interneurons become extensively myelinated very early in development[34], which is thought to be important for preventing runaway excitation in nascent circuits and for coordinating pyramidal neuron firing to generate high-frequency oscillations[35,36]. Recent in vivo-imaging studies indicate that the distribution of oligodendrocytes and the overall pattern of myelin changes upon recovery from cuprizone-induced demyelination[37], but it is not known if this reorganization affects the distribution and content of myelin along excitatory and inhibitory neurons equally. Myelin within layer I of the somatosensory cortex is completely restored after cuprizone-induced destruction of oligodendrocytes[37], providing a unique opportunity to study the specificity of the remyelination process in vivo.

Here we used a combination of cell-specific axon labeling with high-resolution imaging of myelin sheaths to define the myelination patterns of seven distinct excitatory and inhibitory neuronal subtypes that extend axon collaterals within layer I of the somatosensory cortex. This analysis revealed a diversity of myelin patterns that did not closely follow functional class (excitatory versus inhibitory) or cell body location (thalamus versus cortex), with discontinuous myelination observed on all myelinated axons in these adult mice. The probability of myelination could be predicted through a combination of cell type and axon diameter, indicating that the observed patterns are strongly influenced by cell intrinsic factors rather than just cell morphology. By performing longitudinal time-lapse imaging of local PV interneurons and thalamocortical VM neurons we found that although the precise pattern of myelin along individual axons was altered following regeneration, at the population level, the total myelin content on these distinct neuronal subtypes was preserved. These findings suggest that regeneration of oligodendrocytes relies on opportunistic target selection to restore the content of myelin on diverse neuronal subtypes in the mammalian cortex.

## Results

**Cortical axons exhibit diverse myelination patterns.** To define the pattern of myelination along axons of different subtypes of neurons in the cerebral cortex, we used Cre-lox and viral expression strategies to fluorescently label seven neuronal subpopulations that extend axon collaterals into layer I of primary somatosensory cortex (S1) (Fig. 1a, Table 1). We compared myelin along axons of two types of GABAergic neurons (parvalbumin (PV), somatostatin (SOM)) and three corticocortical projecting neurons (layer VI corticothalamic: NTSR1-Cre[38–40]; layer VIb subplate: NXPH4-Cre[41,42]; layer Va/Vb pyramidal: RBP4-Cre[43]) by crossing to fluorescent reporter mice (Ai3, Ai9)[44]. These Cre lines have been well-characterized in previous studies and are highly selective in labeling pyramidal neurons within their corresponding lamina[45,46] (Supplemental Fig. 1a–j). In addition, two thalamocortical projections, the ventral medial nucleus (VM) and medial posterior nucleus (PO, were analyzed by injecting tdTomato expressing AAV virus within different regions of the thalamus (Supplemental Fig. 1k–n). Horizontal sections that preserved the orientation of axonal trajectories (parallel to the pia) were then collected from these mice and immunostained for MBP (Fig. 1b, c). The complete morphologies of randomly selected axons were traced within a 675 µm × 675 µm × 40 µm volume and the percent of axon length myelinated (PLM) was calculated ([total length of all MBP-positive internodes/total axon length] × 100%) (Fig. 1d–l, Supplementary Movie 1). PLM distributions varied considerably between the six neuronal subtypes that were myelinated (Fig. 2a, b); NTSR1-expressing pyramidal neurons were not myelinated (Fig. 2c). Moreover, axons of each neuronal subtype ranged from unmyelinated to nearly completely myelinated, revealing the high variability in cortical myelination even within the same neuronal subtype. Although differences in the degree of axonal collateralization between neuronal subtypes influenced the total axon length measured (Fig. 2d), this variability did not correlate with PLM for any neuronal subtype except SOM ($R^2 = 0.373$) (Supplemental Fig. 1), suggesting that under-sampling did not strongly influence the detection of these myelination patterns.

As previously described[7,8], PV axons were among the most highly myelinated, with approximately two-thirds of traced axons associated with at least one internode (average proportion: $0.7 \pm 0.08$, $n = 4$ mice) and about half of the PV axons exhibiting a PLM > 50% (Fig. 2a–c). The proportion of PV axons with myelin in layer I was lower than previously reported[8]. We expect this apparent discrepancy is due to tracing only the axonal arbor contained in layer I, where the degree of myelination is several times lower than that in layer II/III[7,8]. VM thalamocortical axons were myelinated to a similar extent as PV axons (average

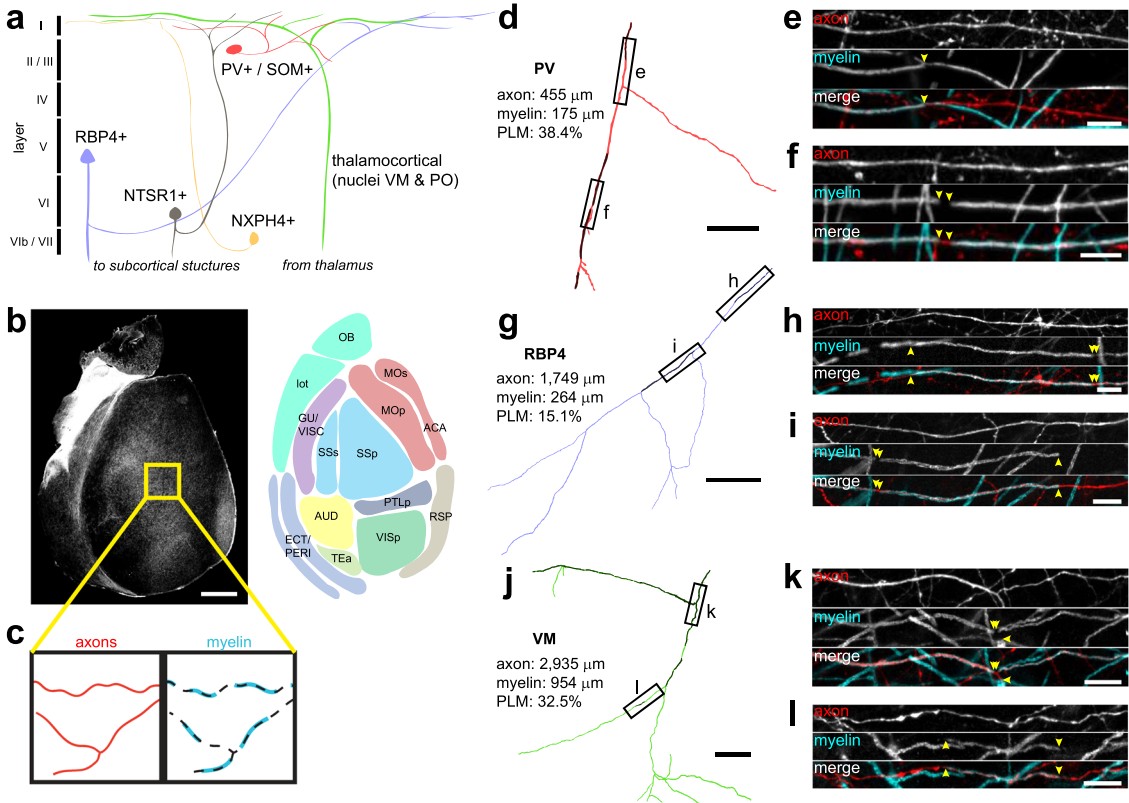

**Fig. 1 High-resolution tracing of distinct axon populations reveals discontinuous myelination patterns. a** Schematic of the cortex indicating the sources of axons from the distinct neuronal populations investigated. Note that the axons depicted may not necessarily be the primary projection of each neuron. **b** A horizontal section of the entire cortical flatmount and schematic representing approximate cortical areas. $N = 4$ mice per neuron subtype. **c** Schematic of experimental design. Traces (black dotted line) of fluorescently labeled axons (red) were used to trace corresponding MBP-immunostained myelin sheaths (cyan). **d–l** Full reconstructions of traced PV (**d**), RBP4 (**g**), and VM (**j**) axons within cortical layer I and their associated dimensions (PLM, percent length myelinated). Black myelin sheaths are overlaid on colored axons. Labeled rectangles highlight example myelinated segments of PV (**e**, **f**), RBP4 (**h**, **i**), and VM (**k**, **l**) axons. Yellow arrowheads indicate the ends of myelin segments (MBP immunoreactivity). $n = 60$ (PV), 62 (VM), 60 (SOM), 61 (PO), 60 (NXPH4), 78 (RBP4), 60 (NTSR1) axons. Scale bars, **b**, 1 mm; **d**, **g**, **j**: 100 µm; **e**, **f**, **h**, **i**, **k**, **l**: 10 µm. OB olfactory bulb, lot lateral olfactory tract, MOp/MOs primary/secondary motor, ACA anterior cingulate area, GU/VISC gustatory/visceral, SSp/SSs primary/secondary somatosensory, PTLp posterior parietal association, AUD auditory, RSP retrosplenial, VISp primary visual, ECT/PERI ectorhinal/perirhinal, TEa temporal association.

**Table 1 Layer I-projecting neuronal subtypes and labeling strategies.**

| Neuronal subtype | Examined in this study? | Labeling strategy | Myelination status |
|---|---|---|---|
| Serotonergic | No | N/A | Not myelinated[85] |
| Adrenergic | No | N/A | Not myelinated[86] |
| Dopaminergic | No | N/A | Not myelinated[87] |
| Cholinergic | No | N/A | Unknown |
| Thalamocortical (ventromedial nucleus) | Yes | AAV-DJ-CaMKii-mCherry or AAV9-hSyn-EGFP | Strongly myelinated |
| Thalamocortical (posterior nucleus) | Yes | AAV-DJ-CaMKii-mCherry | Intermediately myelinated |
| Layer II/III | No | N/A | Unknown |
| Layer V intracortical | Yes | *Rbp4-Cre; Ai9* or AAV9-CAG-FLEX-tdTomato injections into motor cortex of *Rbp4-Cre* mice | Weakly myelinated |
| Layer VI intracortical | Yes | *Ntsr1-Cre; ROSA-lsl-EYFP (Ai3)* or ROSA-lsl-tdTomato (*Ai9*) | Not myelinated |
| Layer VIb subplate | Yes | *Nxph4-CreER; Ai9* | Weakly myelinated |
| Parvalbumin+ interneuron | Yes | *Pvalb-Cre; Ai9* | Strongly myelinated |
| Somatostatin+ interneuron | Yes | *Sst-Cre; Ai9* | Intermediately myelinated |

proportion: $0.6 \pm 0.1$, $n = 4$ mice) and exhibited similar myelination patterns ($p = 0.99$, Kruskal–Wallis one-way ANOVA with Dunn–Šidák correction for multiple comparisons) (Fig. 2a–c, Table 2). All other axons examined were myelinated much less frequently (average proportions: PO, $0.3 \pm 0.06$; SOM, $0.3 \pm 0.05$; RBP4, $0.08 \pm 0.04$; NXPH4, $0.05 \pm 0.03$; NTSR1, 0; $n = 4$ mice each) (Fig. 2c, Table 2). There was no difference in the average length of internodes between neuronal subtypes ($p = 0.514$,

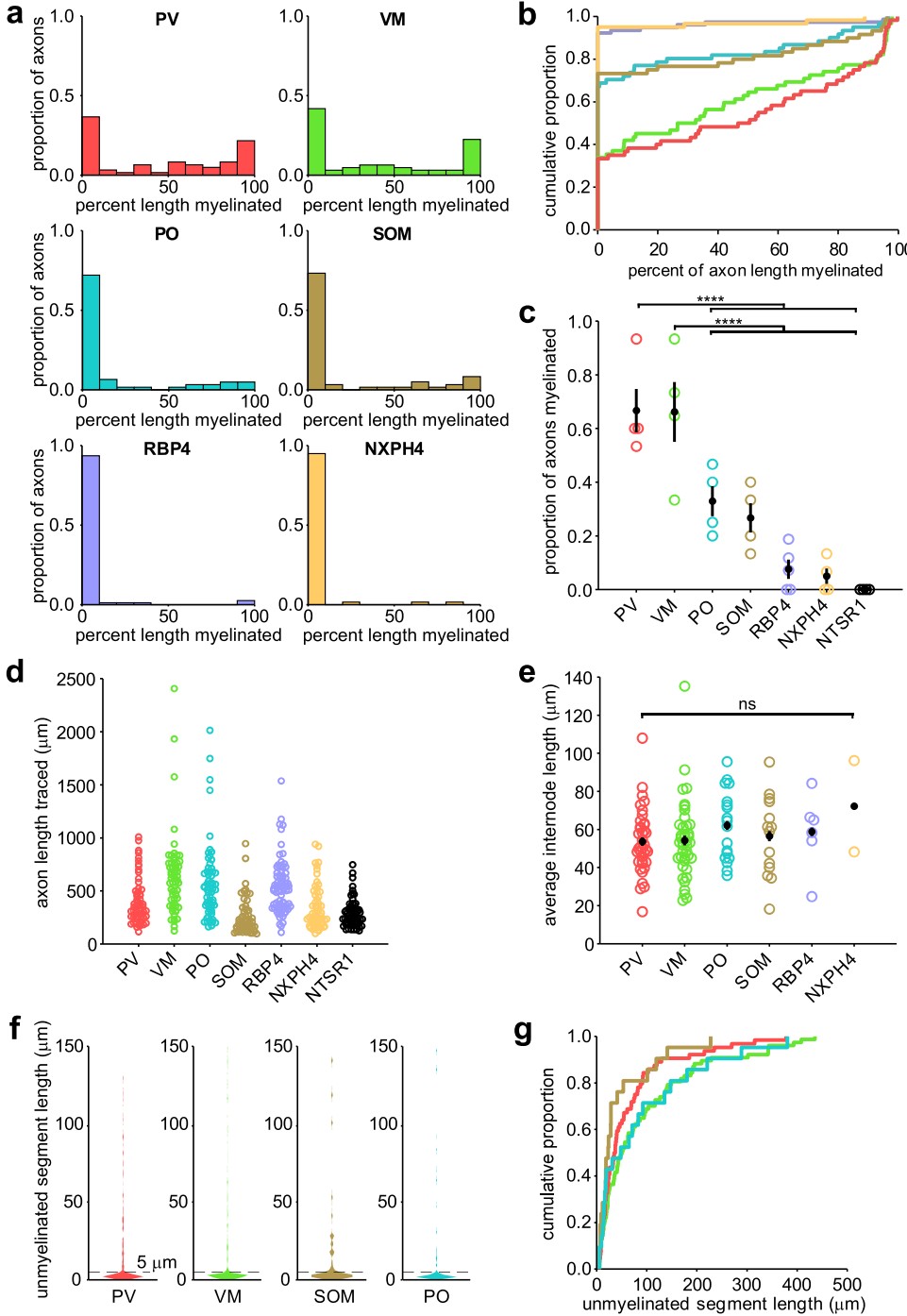

**Fig. 2 Different neuronal subtypes exhibit diverse myelination profiles. a** Myelination profiles for the six neuronal subtypes examined in this study with myelinated axons. Each plot is a histogram of pooled data across four animals for each subtype representing the proportion of axons with different percent length myelinated in 10% bins (n = 60 (PV), 62 (VM), 60 (SOM), 61 (PO), 60 (NXPH4), 78 (RBP4), 60 (NTSR1) axons). See Table 2 for statistical comparisons. **b** Cumulative distribution plot of data in (**a**), showing three distinct classes of myelination patterns. **c** Proportions of axons myelinated per neuronal subtype (N = 4 mice for each subtype) ****p < 0.0001. **d** Lengths for each axon traced (same Ns as in (**a**)). **e** Average internode lengths per axon for each neuronal subtype (n = 38 (PV), 41 (VM), 18 (SOM), 14 (PO), 6 (NXPH4), 2 (NTSR1) axons) (ns, not significant, p = 0.514, one-way ANOVA). **f** Violin plots of the lengths of unmyelinated axon segments (axonal distance between two consecutive MBP segments) for the top four most strongly myelinated neuronal subtypes. Unmyelinated segments of <5 μm (typical maximum length of nodes of Ranvier) are most common, while unmyelinated segments between 5 and 150 μm distributed broadly. **g** Cumulative distribution plot of all non-nodal (>5 μm) unmyelinated segments from data in **f**. All error bars represent the standard error of the mean. Source data and statistical tests are provided as a Source Data file.

one-way ANOVA; Fig. 2e), indicating that PLM differences reflect the number of internodes per axon.

Three general myelination profiles were evident in the PLM distributions: strongly myelinated (PV and VM), intermediately myelinated (SOM and PO), and weakly myelinated (RBP4 and NXPH4) (Fig. 2b). Overall, PO and SOM axons shared similar myelination patterns, matching that observed in deeper cortical layers[34], as did RBP4 and NXPH4 axons. Notably, these

**Table 2 Multiple comparison tests for myelination patterns.**

| Sample 1 | Sample 2 | Lower 95% CI | Estimate | Upper 95% CI | Corrected $p$ value |
|---|---|---|---|---|---|
| PV | VM | −42.98 | 5.88 | 54.74 | 1 |
| PV | SOM | 29.57 | 78.83 | 128.09 | $4.19 \times 10^{-5}$ |
| PV | PO | 23.95 | 73.01 | 122.06 | 0.0002 |
| PV | RBP4 | 71.34 | 117.67 | 164.00 | $1.56 \times 10^{-12}$ |
| PV | NXPH4 | 72.66 | 121.92 | 171.18 | $6.43 \times 10^{-12}$ |
| VM | SOM | 24.09 | 72.95 | 121.81 | 0.0002 |
| VM | PO | 18.47 | 67.12 | 115.78 | 0.0008 |
| VM | RBP4 | 65.88 | 111.79 | 157.69 | $1.51 \times 10^{-11}$ |
| VM | NXPH4 | 67.17 | 116.03 | 164.90 | $5.37 \times 10^{-11}$ |
| SOM | PO | −54.89 | −5.83 | 43.23 | 1 |
| SOM | RBP4 | −7.49 | 38.84 | 85.17 | 0.1921 |
| SOM | NXPH4 | −6.18 | 43.08 | 92.34 | 0.1457 |
| PO | RBP4 | −1.45 | 44.66 | 90.78 | 0.0664 |
| PO | NXPH4 | −0.15 | 48.91 | 97.97 | 0.0514 |
| RBP4 | NXPH4 | −42.08 | 4.25 | 50.58 | 1 |

Results of Kruskal–Wallis one-way ANOVA with Dunn–Šidák correction for multiple comparisons of data in Fig. 2.

myelination classes did not match functional classifications: while all corticocortical axons were very weakly myelinated, the interneuronal and thalamocortical axons were each split between strongly and intermediately myelinated classes.

Myelination along individual cortical axons is often discontinuous, with myelinated segments interrupted by long stretches of unmyelinated axon[9–11]. To determine the extent of myelin discontinuity along different axons, we quantified the lengths of unmyelinated segments among the well-myelinated neuronal subtypes (PV, VM, SOM and PO). To reach the same PLM, axons could be myelinated with frequent gaps between adjacent internodes or have infrequent, longer gaps between local continuously myelinated regions, with the only interruptions arising from nodes of Ranvier. Plots of unmyelinated segment length revealed that all axons exhibited a large cluster of short lengths typical for nodes of Ranvier (0–5 µm; PV: 67%, VM: 73%, PO: 66%, SOM: 69% of all lengths) (Fig. 2f), indicating that internodes tend to be clustered along axons to form continuously myelinated segments. To explore whether there are differences in the distribution of myelin between neuron subtypes, we generated cumulative distributions of non-nodal (>5 µm) unmyelinated segments (Fig. 2g). Although PV and SOM neurons tended to have shorter unmyelinated segments, there were no statistically significant differences between the distributions ($p = 0.095$, Kruskal–Wallis one-way ANOVA). Together, these studies reveal that different neurons that extend axons within layer I of the adult somatosensory cortex exhibit distinct, highly variable myelination patterns.

**Axonal myelination patterns scale between cortical regions.** The density of myelin varies considerably across the cortical mantle (Fig. 1b), with boundaries between cortical areas defined by their distinct myeloarchitecture[47,48]. It is not known if these differences reflect changes in myelin content along with certain classes of neurons or proportional decreases in myelin content along all axons. To determine if the diversity in myelin patterns observed in the somatosensory cortex are preserved in other cortical regions, we compared the myelination of PV axons across five cortical areas, as PV neurons exhibit consistent axon density across the primary sensory and motor cortices, with the exception of a slightly lower density in association cortex[49]. Among sensory cortical areas (somatosensory (SS), visual (VIS), and auditory (AUD)), the density of MBP immunoreactivity was similar (mean gray value of binarized $z$ projection, SS: 50 ± 8; VIS: 66 ± 8; AUD: 61 ± 11, arbitrary units) (Fig. 3a) and the PLM distribution of PV

axons was not significantly different between these areas (SS vs. VIS: $p = 0.16$; SS vs. AUD: $p = 0.45$; VIS vs. AUD: $p = 1$, Kruskal–Wallis one-way ANOVA with Dunn–Šidák correction for multiple comparisons) (Fig. 3b, c, Table 3). As expected, in cortical areas with lower myelin content (i.e. secondary motor (MOs): 42 ± 8; temporal association area (TEa): 14 ± 3), the PLM distribution was significantly left-shifted, with fewer PV axons myelinated (Fig. 3b, c; Table 3). However, even in these sparsely myelinated areas, PV axons were among the most highly myelinated; when the myelin abundance on PV axons was scaled to overall MBP density (scaled myelination prevalence = average binarized MBP intensity/proportion of axons myelinated) this value was not significantly different between cortical regions ($p = 0.24$, Kruskal–Wallis ANOVA) (Fig. 3d), suggesting that PV axon myelination scales with the overall density of myelin. The proportional scaling of myelin content on PV neurons suggests that the high myelination probability of these axons is preserved between cortical regions and that variations in myelin abundance primarily arise through regional differences in oligodendrogenesis or oligodendrocyte survival. Performing a similar global analysis on other neuron subtypes is complicated by the diverse populations of axons that reside in distinct regions of the cortex, so it is not yet known if myelination probabilities are conserved for all neuronal subtypes.

**Axon diameter and neuronal subtype influence myelination patterns.** The relationship between axonal diameter and myelination has been thoroughly described in the context of peripheral nerves, CNS white matter, and even primary cell culture[12–17,50], establishing that the probability of myelination increases with axonal diameter. To determine if differences in myelination between neuronal subtypes in the cortex are simply due to differences in axon diameter, we quantified the variation in axon diameter among each neuronal subtype in relation to MBP + myelin sheaths (Fig. 4a). Axon diameters were estimated by measuring the full-width at half maximum (FWHM) of fluorescent intensity (EYFP, tdTomato, or mCherry) across axons using superresolution microscopy (Fig. 4b). Across all neuronal subtypes examined there was a positive correlation between axon diameter and probability of being myelinated ($p = 8.99 \times 10^{-14}$, two-sample Kolmogorov–Smirnov test) (Fig. 4c, d). However, for axons with diameters between 0.4 and 0.8 µm (44% of all axons examined) there was remarkable diversity in myelination status among all neuron subtypes, varying from completely myelinated to completely unmyelinated (Fig. 4c, d). While the most strongly

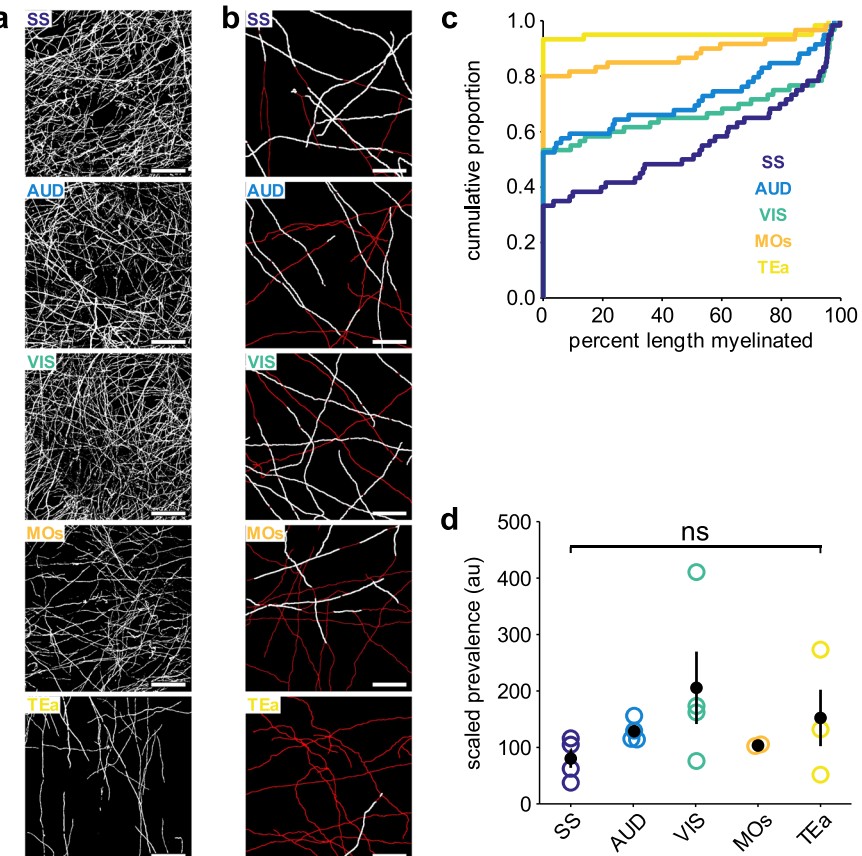

**Fig. 3 PV axon myelination patterns are consistent across cortical regions. a** Representative binarized images of MBP immunostaining from all cortical regions analyzed (SS somatosensory, AUD auditory, VIS visual, MOs secondary motor, TEa temporal association area). Scale bars, 50 μm. **b** Example axon and myelin traces from each cortical region analyzed from a single *PV-Cre; Ai9* flatmount. Red, axons; white, myelin. Scale bars, 50 μm. **c** Cumulative distribution plot of PLM from all axons traced in each region in **a** (SS, *n* = 60 axons from 4 mice; VIS, *n* = 60 axons from 4 mice; AUD, *n* = 59 axons from 4 mice; MOs, *N* = 30 axons from 2 mice; TEa, *N* = 37 axons from 3 mice). See Table 3 for PLM distribution statistics. **d** Scaled prevalence (MBP intensity/proportion of axons myelinated) of PV axon myelination across all regions analyzed (same *N*s as in **c**) (ns not significant, *p* = 0.24, Kruskal–Wallis ANOVA). Error bars represent the standard error of the mean. Source data and statistical tests are provided as a Source Data file.

**Table 3 Multiple comparison tests for PV regional myelination.**

| Sample 1 | Sample 2 | Lower 95% CI | Estimate | Upper 95% CI | Corrected *p* value |
|---|---|---|---|---|---|
| SS | VIS | −5.54 | 33.03 | 71.60 | 0.1534 |
| SS | AUD | −11.73 | 26.68 | 65.09 | 0.4124 |
| SS | MOs | 33.91 | 72.32 | 110.73 | $1.36 \times 10^{-6}$ |
| SS | Tea | 51.46 | 89.87 | 128.28 | $5.74 \times 10^{-10}$ |
| VIS | AUD | −44.91 | −6.34 | 32.23 | 1 |
| VIS | MOs | 0.72 | 39.29 | 77.86 | 0.0426 |
| VIS | Tea | 18.27 | 56.84 | 95.41 | 0.0004 |
| AUD | MOs | 7.22 | 45.63 | 84.04 | 0.0088 |
| AUD | Tea | 24.77 | 63.18 | 101.59 | $4.11 \times 10^{-5}$ |
| MOs | Tea | −20.86 | 17.55 | 55.96 | 0.8937 |

Results of Kruskal–Wallis one-way ANOVA with Dunn–Šidák correction for multiple comparisons of data in Fig. 3.

myelinated subtypes (PV and VM) tended to have larger axon diameters (average axon diameter: PV, 0.6 ± 0.03 μm; VM, 0.5 ± 0.03 μm), their thinnest segments (<0.5 μm) still had a reasonable likelihood of being myelinated (PV, 30%; VM, 21%) (Fig. 4c). To explore relationships between cell identity, axon diameter, and myelination status, we fitted a binomial generalized linear model to the entire dataset, whereby the probability of an axon being myelinated is predicted by axon diameter and neuronal subtype (P(myelinated) ~ 1 + diameter + subtype) (Table 4). This analysis

revealed that while axon diameter influences myelination probability (Fig. 4e), neuronal subtype also effects myelination status, visible in the leftward shift in probability curves for neurons with highly myelinated axons (e.g. PV and VM), reaching 50% probability of myelination at just over 0.5 μm (prediction [95% confidence intervals] = VM: 0.5 [0.31, 0.69] μm; PV: 0.5 [0.30, 0.70] μm). Myelin probability curves for the remaining neuron subtypes were right-shifted, reaching 50% probability of myelination at 0.6 μm [0.25, 0.74] (PO), 0.7 μm [0.21, 0.79] (RBP4),

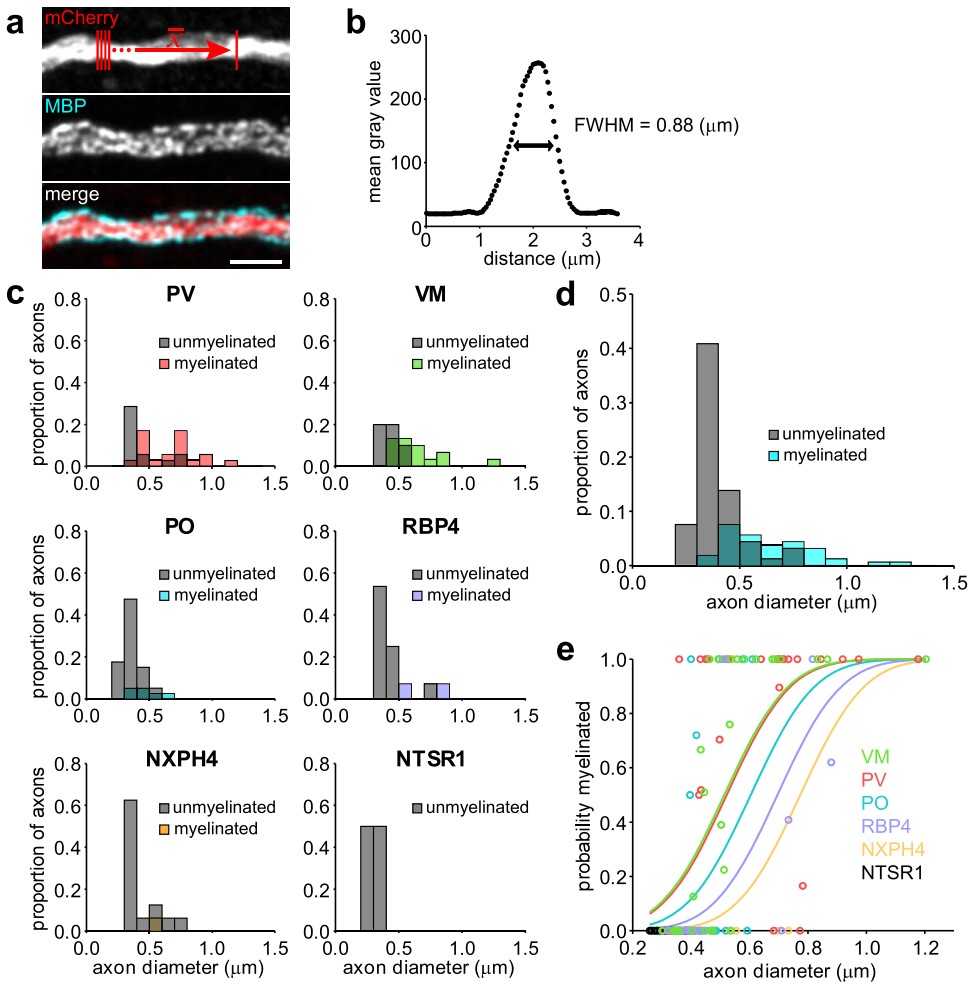

**Fig. 4 A combination of diameter and neuronal subtype predicts myelination status. a** Example segment of an axon imaged at high resolution for diameter analysis and corresponding MBP immunostaining. A small rectangle was used to plot a profile of the average gray value intensity profile across a few μm of the axon (result shown in (**b**)) to determine its diameter by the full width at half maximum (FWHM). Scale bar, 2 μm. **b** Plot of the average profile across the axon in (**a**). Each point represents the average gray value of a single pixel across the representative lines in (**a**), top panel. **c** Histograms of the proportions of axons myelinated (colored bars) or unmyelinated (gray bars) for different axon diameters. Note the color of overlapping bars is additive. **d** As in (**c**), pooled across all neuronal subtypes. **e** Probability distributions for the myelination probability of each neuronal subtype given a certain diameter. Points are individual axons myelinated. Intermediate probabilities (not 1 or 0) represent the average myelination status across at least three positions along the axon surveyed to calculate the average diameter. See text for details on probability functions. Source data and statistical tests are provided as a Source Data file.

**Table 4 Estimated coefficients for the generalized linear model.**

|  | Estimate | SE | t stat | p value |
|---|---|---|---|---|
| (Intercept) | −3.044 | 0.581 | −5.237 | $1.63 \times 10^{-7}$ |
| diam | 5.836 | 1.034 | 5.646 | $1.64 \times 10^{-8}$ |
| type_NTSR1 | −15.215 | 2573735.335 | 0.000 | 1.000 |
| type_NXPH4 | −1.470 | 0.650 | −2.262 | 0.024 |
| type_RBP4 | −1.014 | 0.457 | −2.219 | 0.026 |
| type_PO | −0.481 | 0.389 | −1.235 | 0.217 |
| type_VM | 0.047 | 0.367 | 0.127 | 0.899 |

Estimated coefficients for the binomial generalized linear model of data in Fig. 4, whereby the probability of an axon being myelinated is predicted by axon diameter and neuronal subtype (P (myelinated) ~ 1 + diameter + subtype). *Note*: Coefficient estimates and *p* values are relative to PV.

and 0.8 μm [0.11, 0.89] (NXPH4) (Fig. 4e). Together, these results indicate that myelination of cortical axons does not follow a strict diameter–myelination relationship similar to that described in the PNS[12–14], and that other neuron intrinsic factors

profoundly influence the probability of being selected for myelination.

**Remyelination restores myelin patterns among different neuronal subtypes.** Recent studies indicate that oligodendrocyte regeneration from endogenous progenitors is sufficient to restore myelin levels in layer I after demyelination; however, the pattern of myelin along individual axons is altered after remyelination, with a large fraction of sheaths established on previously unmyelinated axon segments[37,51]. It is not yet known if this reorganization is equally distributed across different neuronal subtypes. If the cell-intrinsic features that shape myelination during development are altered by demyelination, it could result in profound alterations in circuit properties, despite restoring myelin levels. To address this question, we used longitudinal in vivo imaging to compare the myelination patterns of individual PV and VM axons before and after remyelination, as they have the highest probability of being myelinated and the largest variation in myelination patterns (Figs. 1a, b, and 4e). To achieve simultaneous, two-color labeling of PV and VM axons in vivo,

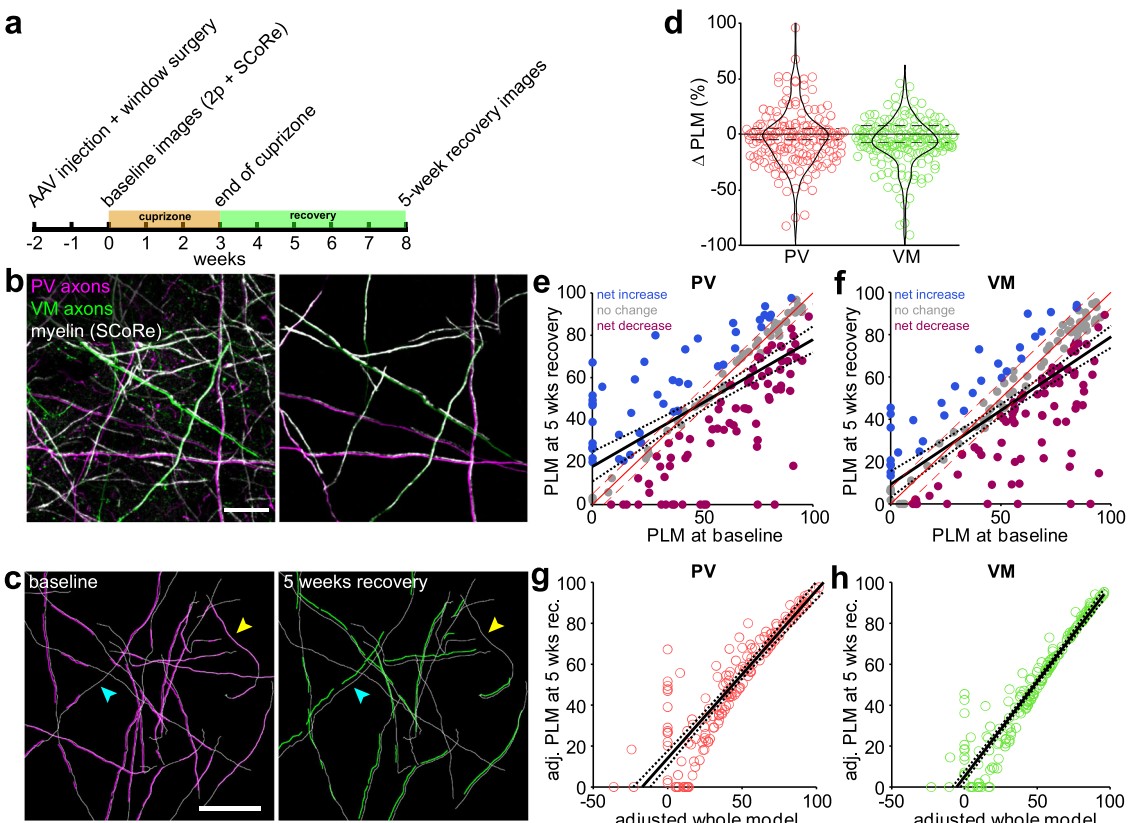

**Fig. 5 Overall subtype-specific myelination patterns are restored despite remodeling of individual axons. a** Experimental timeline. **b** Example image from a *PV-Cre; Ai9* mouse injected with AAV-EGFP into thalamic VM. SCoRe signal from the same region is overlaid on top of the fluorescent dual-colored image acquired with two-photon microscopy. Right panel shows a few myelinated axons of both neuronal subtypes extracted from the left panel. Scale bar, 30 μm. **c** Axon traces from an example 3-week cuprizone timecourse. Left panel shows PV axon traces (white) and the associated myelin segments traced from the corresponding SCoRe image (magenta). Right panel shows same region with the surviving baseline axon traces overlaid with the positions of SCoRe segments traced at 5 weeks of recovery (green). Yellow arrowhead indicates example axon that had a net loss in length myelinated, while the cyan arrowhead indicates a portion of axon which gained myelination. Scale bar, 100 μm. **d**, Pooled data from remyelinated PV and VM axons showing the net change in PLM between baseline and 5 weeks recovery where −100% represents an axon going from fully myelinated to fully unmyelinated, and +100% represents an axon that went from unmyelinated to fully myelinated. Dashed lines bordering 0% (solid line) reflect the tracing error as calculated by the average negative ΔPLM from axons of the respective subtype traced in control animals (Supplemental Fig. 2, see text for additional details). **e** and **f** Scatter plots showing the relationship between the PLM of an axon at baseline and its PLM at five weeks recovery from cuprizone for PV axons (**e**) and VM axons (**f**). The solid red line ($y = x$) is the identity line (no change between baseline and 5 weeks recovery) and the red dashed lines are the error based on control traces (as in (**d**)). The black line is the best fit found by regression and the black dotted lines are the 95% confidence bounds. **g** and **h** Data as plotted in (**e**, **f**) readjusted to include the ΔPLM of each axon in the linear model (adjusted formula: PLM at 5 weeks recovery ~ 1 + baseline PLM + baseline PLM: ΔPLM; **g**, PV axons; **h** VM axons). The solid line is the fit and dotted lines are the 95% confidence bounds. Source data and statistical tests are provided as a Source Data file.

AAV encoding EGFP was injected into the VM nucleus of the thalamus of *PV-Cre;Ai9* mice. Cranial windows were then installed over the somatosensory barrel field before mice underwent a 3-week cuprizone protocol shown previously to almost entirely demyelinate the superficial layers of cortex[37,52] (Fig. 5a). Two photon imaging was used to record the distribution of PV and VM axons in layer I and spectral confocal reflectance (SCoRe) microscopy[53] was used to identify myelin sheaths in the same field of view (Fig. 5b). Comparison of myelin patterns before and after oligodendrocyte regeneration using this approach revealed that the majority of axons of both neuronal subtypes were myelinated before cuprizone exposure and remyelinated after 5 weeks of recovery (PV: 140/197 axons; VM: 150/214 axons; $n = 12$ regions from 9 mice) (Fig. 5c), a time point at which myelin density is fully restored[37] (Supplemental Fig. 3). This result indicated that the high preference for myelinating these axons is preserved after demyelination.

To determine if remyelination alters the distribution of myelin along with individual PV and VM axons, each myelin internode was traced along each labeled axon within the imaging volume before cuprizone and after recovery. During this timeframe, there were no large-scale alterations in axonal morphology such as branching, which can strongly impact myelination[54]. Our analysis revealed that although the percent change in axon length myelinated (ΔPLM) for individual axons varied considerably (Fig. 5c), there was little change in average PLM across the population of axons for either PV or VM neurons (PV: −3.7 ± 1.9%, VM: −7.7 ± 1.6%) (Fig. 5d). Together, these results indicate that while the pattern of myelination along individual axons in the cortex changes after a demyelinating event, the overall myelin content on PV and VM axons is restored by remyelination.

To define the relationship between baseline and recovery myelin patterns, we plotted the PLM for individual PV and VM axons at five weeks recovery versus baseline and fit a linear model

to these distributions (Fig. 5e, f). There was a strong positive correlation between baseline PLM and recovery PLM for individual axons (PV: $R^2 = 0.389$, $p = 1. \ 01 \times 10^{-19}$; VM: $R^2 = 0.531$, $p = 2.86 \times 10^{-30}$), with minimal bias from differences in axon length traced (PV: $R^2 = 0.049$, VM: $R^2 = 0.019$) (Fig. 5e, f). Most values in these plots were close to the identity line (0% ΔPLM) at PLM values > 60% (solid red lines, Fig. 5e, f), in accordance with prior in vivo studies indicating that continuously myelinated axons are the most likely to be remyelinated[37,51]. However, for both PV and VM neurons there were more well-myelinated axons below the identity line, suggesting that there was a slight trend toward loss of myelin after recovery (ΔPLM for axons with >75% PLM at baseline, PV: median: −6%, range: [−82%, 15%]; VM: median: −8%, range: [−90%, 10%]) (Fig. 5e, f). Conversely, many unmyelinated or weakly myelinated axons with low PLM at baseline had large gains in myelination during recovery (ΔPLM for axons with <25% PLM at baseline: PV, median: 21%, range: [−20%, 95%]; VM, median: 6%, range: [−14%, 46%]) (Fig. 5e, f). As a result, the best fit lines (solid black lines) cross the identity line at 45% and 31% for PV and VM axons, respectively, rather than 50% if myelin content before and after regeneration were identical. These data suggest that weakly myelinated axons are more likely to gain myelin during recovery, while continuously myelinated axons are more likely to lose myelin. To test this hypothesis, we refitted the linear model to test for an interaction between PLM at baseline and the ΔPLM exhibited during recovery (recovery PLM ~ 1 + baseline PLM + baseline PLM: ΔPLM). After adjusting for this preference, the updated model strongly predicted PLM outcomes at 5 weeks recovery for both PV and VM axons (PV: $R^2 = 0.773$, $p = 3.06 \times 10^{-54}$; VM: $R^2 = 0.907$, $p = 2.87 \times 10^{-89}$) (Fig. 5f, g), supporting the hypothesis that regeneration induces a slight shift in myelination extent from well-myelinated to sparsely myelinated axon segments of both PV and VM neurons.

**PV axon remyelination remains proportional in regions with lower oligodendrocyte regeneration.** The extent of oligodendrocyte regeneration after cuprizone exposure varies dramatically between cortical layers; after two months of recovery, oligodendrocytes are restored to their original density in layer I, but reach only ~50% of their original density in layer II/III[37]. If axon selection by oligodendrocytes during regeneration follows rules similar to that observed in the naïve brain, then remyelination of PV axons in these deeper cortical layers should remain proportional to the overall myelin density. Alternatively, PV axons may be preferentially remyelinated when oligodendrogenesis is suppressed to rapidly restore inhibition. To determine the extent of PV axon remyelination in layer II/III, a genetic labeling approach was used to visualize myelin rather than SCoRe imaging, because light is only reflected by myelin that is approximately perpendicular to the incident light[53]. While SCoRe imaging is effective for visualizing horizontal sheaths in layer I, layer II/III contains mostly vertically oriented sheaths. By crossing *Mobp-EGFP* mice to *PV-Cre; Ai9* mice, myelin around individual PV axons could be resolved in vivo using two-photon imaging (Fig. 6a, Supplemental Movie 2). We performed repetitive imaging through a chronic cranial window to follow individual myelin sheaths in layer II/III through cuprizone-induced demyelination and subsequent 5 weeks of recovery. Individual myelin sheaths were subsequently traced within randomly selected 100 μm × 100 μm × 100 μm ROIs and identified as wrapping labeled PV axons or unlabeled non-PV ("other") axons (Fig. 6b). Although the total number of sheaths within an ROI was lower than baseline after five weeks, as expected for incomplete remyelination (baseline: 156 ± 12 sheaths; 5 weeks recovery: 109 ± 13 sheaths; $p = 0.003$, paired $t$-test) (Fig. 6b, c), the percentage of all myelin surrounding PV axons was comparable to that observed at baseline (baseline: 45 ± 2%; 5 weeks recovery: 48 ± 2%; $p = 0.068$, paired $t$-test) (Fig. 6d). These results indicate that PV axons are selected for

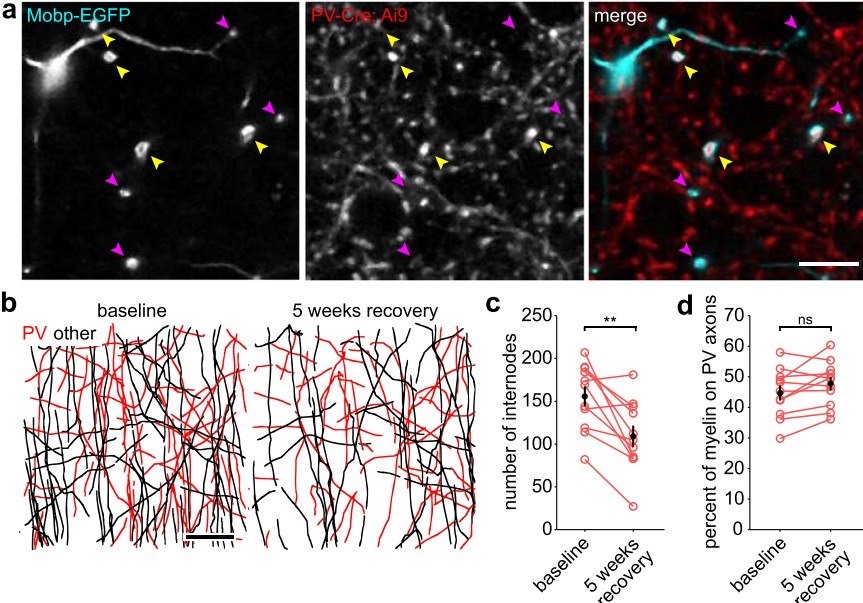

**Fig. 6 PV axons are not preferentially remyelinated in regions with reduced oligodendrogenesis. a** In vivo image of layer II/III somatosensory cortex acquired from a *Mobp-EGFP; PV-Cre; Ai9* mouse. Yellow arrowheads identify example sheaths surrounding PV axons, while magenta arrowheads identify sheaths surrounding non-PV axons ("other"). Scale bar, 10 μm. **b** Example maximum *XZ* projection of an ROI (100 μm × 100 μm × 100 μm) from layer II/III showing all traced internodes within the volume at baseline (left) and at 5 weeks recovery from cuprizone (right). Red, sheaths surrounding PV axons; black, sheaths surrounding "other" axons. Scale bar, 20 μm. **c** Total number of internodes within an ROI at baseline vs. 5 weeks recovery ($n = 10$ ROIs (up to 2 ROIs from 2 imaged regions across 3 mice)) (**$p = 0.003$, paired two-tailed $t$-test). **d** Percent of traced sheaths surrounding PV axons within an ROI at baseline vs. 5 weeks recovery, same $N$s as in **c** (ns, not significant, $p = 0.068$, paired two-tailed $t$-test). All error bars represent standard error of the mean. Source data and statistical tests are provided as a Source Data file.

remyelination with the same probability, independent of the amount of myelin that is available, providing further evidence that target selection by oligodendrocytes is opportunistic and governed by axon properties.

## Discussion

The cerebral cortex contains axons that arise from the local collaterals and long-range projections of a diverse population of neurons. Despite the presence of axons of suitable size and the persistence of oligodendrocyte progenitors, myelin density in the cortex remains low. The diverse patterns of myelin observed along cortical axons, even among axons from distinct neuron subtypes, raise many new questions about the mechanisms that establish these patterns and the impact of this complex organization on myelin repair. Our in vivo results reaffirm that axonal diameter is a key factor in determining myelination status in this region, as neurons that had the largest axon diameters also had the highest myelin content, recapitulating in vitro studies showing that cortical oligodendrocytes rarely form wraps around structures <0.4 μm[15,55], and the wealth of anatomical studies in the CNS showing that thinner axons are less likely to be myelinated. Nevertheless, many PV and VM axons of relatively thin diameter (<0.5 μm) were continuously myelinated in layer I of the somatosensory cortex, and many large diameter axons (>0.5 μm) were unmyelinated. Moreover, unmyelinated segments often occurred immediately adjacent to myelinated regions along axons with a consistent diameter (Fig. 2f). The lower rate of oligodrogenesis in the cortex relative to white matter[56,57], and the strict, cell-intrinsic limits on myelin output by individual oligodendrocytes[15,19,37], may allow such discontinuous patterns to persist despite the high preference of these axons for myelination. Although oligodendrogenesis and myelination continue in the mouse cortex for many months after birth[9,10,56–58], the rate of oligodendrogenesis in these 5-month-old mice is very low (<1 new oligodendrocyte per month within the imaging volume)[37,56,57,59], and individual myelin sheaths are highly stable[37], indicating that the diverse myelination patterns observed on these distinct excitatory and inhibitory neurons are compatible with the advanced processing capabilities of cortical circuits in the adult brain.

**Regulation of cortical axon myelination**. Although axon diameter is a key determinant of myelination probability, recent evidence suggests that target selection by oligodendrocytes is also influenced by the frequency of axon collateral formation[50], with extensively branched terminal arbors less likely to be myelinated; thus, differences in axonal branching may also contribute to the range of myelination profiles observed in layer I. Consistent with this hypothesis, thalamocortical PO axons that exhibit more terminal arborizations in the somatosensory cortex were less likely to be myelinated for a given axon diameter than VM axons (Fig. 4c), which have more long-range projections that "pass-through" the somatosensory region with limited branching (Supplemental Fig. 1k–n).

In addition to these morphological constraints, our results reveal that neurons exert cell-autonomous control of myelination independent of the diameter and degree of branching of their axons. This finding is consistent with the observation that oligodendrocytes do not form myelin on dendrites, cell bodies, or capillaries, structures that by size alone should be permissive targets. At present, we know very little about the molecular mechanisms that regulate target selection by oligodendrocytes. Several integral membrane proteins that participate in cell adhesion and intercellular signaling can both promote (e.g. Nrg1-III, EphA/B) or prevent (e.g. EphA/B, JAM2, LINGO-1)

myelination[25,60–63], suggesting that these axon–oligodendrocyte interactions could bias this selection process by regulating extension of lamellar sheaths from nascent oligodendrocyte processes[60,64,65], with the relative amounts of these proteins accounting for the bias in subtype remyelination observed. As neuronal activity can promote both oligodendrogenesis and myelination of some neurons[20–22,66–68], it is possible that activity adjusts myelination probability by altering the expression, trafficking, or localization of these proteins. Indeed, a recent study reported that life experience can specifically modulate the myelination of PV axons without affecting myelination of nearby excitatory axons myelinated by the same oligodendrocytes[69].

The vast territory available for the addition of new myelin on axons that have a high probability of being myelinated (e.g. large diameter, low branching) and the persistence of oligodendrocyte progenitors creates opportunities for circuit modification in adulthood by increasing myelin content, a phenomenon termed adaptive myelination. Indeed, motor learning, exposure to new environments, and other life experiences correlate with changes in oligodendrogenesis, myelination, and altered network activity[70]. However, the mechanisms underlying myelin plasticity are unclear and the impact of these changes on circuit behavior remains largely theoretical[71]. In vivo-imaging studies have revealed that less than half of newly formed oligodendrocytes in the cortex become stably integrated[72], suggesting that shifts in myelin content could be achieved by simply regulating integration probability rather than promoting oligodendrogenesis. It is not yet known what types of neurons increase their myelin content in these contexts, whether such changes are long lasting or how such changes ultimately influence the properties of cortical circuits. It is also unclear what adaptive advantage is provided by the gradual addition of myelin during adulthood. The ability to monitor the myelination patterns along individual axons of genetically distinct neuronal subtypes using longitudinal in vivo imaging may provide the means to answer these questions.

**Restoration of cortical myelin through oligodendrocyte regeneration**. The goal of regenerative processes is to restore function. The extraordinary stability of individual myelin sheaths, in terms of their position and length along cortical axons, suggests that these patterns are important for axonal output and thus overall circuit function. However, prior studies demonstrated that overall myelin patterns change in the cortex after destruction and regeneration of oligodendrocytes[37,51,52], and we show here that the position of internodes and total myelin content along individual axons of both local inhibitory PV interneurons and long-range excitatory VM thalamocortical neurons change substantially after remyelination. Nevertheless, following the loss and regeneration of myelin sheaths along individual axons of defined neuron subtypes in vivo revealed that the net change myelin content (ΔPLM) across both populations of axons was negligible after regeneration, despite the considerable reorganization of myelin along with individual axon segments. These findings indicate that the preference for myelinating these axons is preserved after demyelination and raise the possibility that oligodendrocyte regeneration is optimized to restore overall myelin content across the population of diverse axons, rather than the precise patterns of myelin along individual axons. This hypothesis is consistent with the opportunistic behavior of oligodendrocytes, in which axonal selection is determined by a hierarchy of signals (axon diameter, axonal branching, cell identity, activity) that define the myelination probability of an individual axon. As oligodendrocytes are regenerated in new locations[37], reorganization of exact myelin patterns is precluded, but overall myelin content

can be preserved. It is important to note that many of the axons traced in this study extended outside of the bounds of the imaged volumes. Thus, it is possible that the axons with large changes in PLM experienced equal and opposite changes outside the area examined, resulting in precise restoration of myelin content (though not position) along individual axons. New advances that allow imaging over larger areas of the cortex[73] and at greater depths (three-photon excitation) may enable myelin to be visualized on individual axons across several millimeters of the cortex to resolve this question.

PV and VM axons are the most highly myelinated axons investigated in this study; however, due to the high density and vast diversity of axons in layer I, the myelin on these two axon populations represents less than half of the myelin in this region. While we did not determine the remyelination efficiency of other less myelinated neuronal subtypes, similar outcomes are likely. Previous remyelination studies in layer I revealed that isolated sheaths on discontinuously myelinated axons were rarely replaced[37,51] and the number of isolated sheaths lost was approximately the same as those gained on previously unmyelinated axon segments[37]. If myelination patterns were strongly altered in these weakly myelinated axon populations, we would have expected the net ΔPLM to be significantly below zero, as myelin would be "transferred" to other weakly myelinated populations by regenerated oligodendrocytes.

PV axon myelination is highly prioritized in layer II/III, accounting for almost half of all myelin in this region[7,8]. Notably, this preference for PV myelination is in the presence of all the well-myelinated subtypes of layer I (e.g. ascending VM and PO axons). However, layer II/III is poorly remyelinated in comparison to layer I. If the remyelination program prioritized the recovery of PV functionality, one would expect the PV myelin content in layer II/III to increase relative to the total amount of myelin replaced. Our experiments revealed that PV myelin content scaled proportionally with the extent of myelin regeneration. Thus, while myelination of PV axons in layer I was fully restored by 5 weeks recovery, in layer II/III these axons remained poorly myelinated.

The functional implications of these changes in myelination patterns in the cerebral cortex are not well described. Cuprizone-induced demyelination has been shown to cause pyramidal neuron hyperexcitability by shifting axon initial segment position, causing slowed, continuous, rather than saltatory conduction of action potentials and increased conduction failure[74,75]. Moreover, inflammatory myelin loss (EAE, experimental autoimmune encephalomyelitis) results in a dramatic loss of excitatory synapses in both mouse and primate[76]. Hence, alterations in cortical myelin patterns could not only alter action potential conduction and network oscillations but also neuronal connectivity and synaptic plasticity. However, neural circuits are capable of extraordinary adaptation and it is possible that restoring the overall myelin content across a population of axons, without precise replacement of individual internodes, is sufficient to restore circuit function. This possibility provides hope for therapeutic strategies for treating MS based on global promotion of oligodendrogenesis[52,77–79], as the intrinsic properties of axons and the opportunistic behavior of oligodendrocytes may help reestablish the distinct myelin profiles of different neurons in the cortex.

## Methods

**Animal care and use.** Female and male adult mice were used for experiments and randomly assigned to experimental groups. All mice were healthy and did not display any overt behavioral phenotypes. Mice were maintained on a 12-h light/dark cycle, housed in groups no larger than 5, and food and water were provided ad libitum (except during cuprizone-administration, see below). All animal experiments were performed in strict accordance with protocols approved by the Animal Care and Use Committee at Johns Hopkins University. The following transgenic mouse lines were used in this study:

Pvalb-IRES-Cre*[80]
Sst-IRES-Cre[80]
GN220-Ntsr1-Cre[45]
Rbp4-KL100[45]
Nxph4-2A-CreERT2-D[46]
Rosa-CAG-LSL-tdTomato-WPRE (Ai9)*[44]
Rosa-CAG-LSL-EYFP-WPRE (Ai3)[44]
Mobp-EGFP*[81]
*note: C57BL/6 congenic strains were used for cuprizone experiments[81]

For axon tracing in immunostained flatmounts, mice were aged to 5 months, when cortical myelination has nearly plateaued. For in vivo-imaging experiments mice were 10–12 weeks of age when baseline images were acquired.

**CreER induction.** To induce tdTomato expression in the NXPH4-CreER mice, 5-month-old mice were injected with 4-hydroxytamoxifen (4-HT, Sigma) dissolved in sunflower seed oil (Sigma) and administered by i.p. injection twice daily for 5 days at a dose of 100 mg/kg body weight. Mice were perfused 2 weeks from the last day of tamoxifen injection.

**Viral injections.** Five-month-old mice were anesthetized with isoflurane (induction, 5%, mixed with 1 L/min O$_2$; maintenance, 1.5–2%, mixed with 0.5 L/min O$_2$), and their scalps shaved with an electric trimmer. Mice were transferred to a homeostatic heating pad set to 37 °C and their heads secured by ear bars on an Angle 2 stereotax. An incision was performed along the midline of the scalp to expose the skull. Bregma was identified and zeroed in the Angle 2 software. The skull was leveled by ensuring z measurements 2 mm left and right from bregma differed by <0.01 mm and the difference in z between bregma and lambda was <0.1 mm. A small craniotomy was drilled directly above the target coordinates for the injection (VM: −1.45 mm AP, 0.80 mm ML, −4.25 mm DV; PO: −1.35 mm AP, 1.23 mm ML, −3.30 mm DV; MOp: 1.35 mm AP, 1.5 mm ML, −1.83 mm DV) and a micropipette filled with a virus (AAV-DJ-CaMKii-mCherry, $1.2 \times 10^{13}$, or AAV9-hSyn-EGFP-WPRE-bGH, at a titer of $4.44 \times 10^{11}$, or AAV9-CAG-FLEX-tdTomato-WPRE-bGH, at a titer of $1.55 \times 10^{11}$) was slowly lowered to the injection site. The virus was delivered in 13.8 nL pulses over the course of 15 min for a total volume of 150–250 nL. After a 5–10 min wait period to allow diffusion, the pipette was slowly withdrawn, the craniotomy sealed with VetBond, and the scalp closed with metal clamps. For subsequent histological preparation, mice recovered for two weeks to allow sufficient expression and then perfused. For subsequent in vivo imaging, cranial windows were installed immediately following the injection (best window clearing rates) or four to five days later (reduced window clearing rates).

**Flatmount preparation.** Mice were deeply anesthetized with sodium pentobarbital (100 mg/kg w/w) and perfused transcardially first with 20–25 mL warm (~30 °C) PBS (pH 7.4) followed by 20–25 mL ice-cold 4% paraformaldehyde (PFA in 0.1 M phosphate buffer, pH 7.4). Brain hemispheres not used for flatmounts were then post-fixed in 4% PFA for 12 h, before being transferred to a 30% sucrose solution (in PBS, pH 7.4). To generate cortical flatmounts, cortical mantles were carefully dissected from underlying structures directly after perfusion and pressed between glass slides separated by ~1 mm. Cortices were postfixed in this position in 4% PFA for 6–12 h at 4 °C before being unclamped from the glass slides and transferred to 30% sucrose solution. Tissue was stored at 4 °C for more than 48 h before sectioning. Brain tissue was frozen in TissueTek and sectioned at 35–50 µm thickness on a cryostat (Thermo Scientific Microm HM 550) at −20 °C. For flatmount sections, extreme care was taken to ensure the cutting plane was perfectly horizontal to the flatmount surface in order to ensure layer I was contained within the section. First, a mounting chuck was covered with TissueTek and frozen. The mound of TissueTek was then cut until a flat surface across the diameter of the chuck was achieved. Flatmounts were placed pia-down onto a silanized glass slide (hydrophobicity helps prevent tissue sticking), covered with TissueTek, and flipped onto the pre-cut mound. After the tissue was completely frozen, the glass slide was removed, the edges of the TissueTek mound were beveled, and the chuck was mounted for sectioning.

**Immunohistochemistry.** Immunohistochemistry was performed on free-floating sections. Sections were preincubated in blocking solution (5% normal donkey serum, 0.3% Triton X-100 in PBS, pH 7.4) for 1 or 2 h at room temperature, then incubated for 24–48 h at 4 °C or room temperature in primary antibody (listed in Table 5). Secondary antibody (see Table 5) incubation was performed at room temperature for 2–4 h or overnight at 4 °C. Sections were mounted on slides with Aqua Polymount (Polysciences).

**Cranial windows.** Cranial windows were prepared as previously described[37]. Briefly, mice 8–10 weeks old were anesthetized with isoflurane (induction, 5%, mixed with 1 L/min O$_2$; maintenance, 1.5–2%, mixed with 0.5 L/min O$_2$), and their body temperature was maintained at 37 °C with a thermostat-controlled heating plate. The scalp over the right hemisphere was removed and the skull was cleaned and dried. A custom metal head plate with a central hole was attached to the skull

**Table 5 Key resources.**

**Primary antibodies**

| Target Protein/markers | Host species | Source | Catalog # | Identifier | Additional information |
|---|---|---|---|---|---|
| GFP | Chicken | Aves Lab | GFP-1020 | RRID:AB_2307313 | 1:4000 |
| mCherry | Goat | SicGen | AB0040 | RRID:AB_2333092 | 1:5000 |
| MBP | Mouse | Sternberger | 808401 | RRID:AB_2564741 | 1:2000 |
| MBP | Chicken | Aves Lab | F-1005 | RRID:AB_2313550 | 1:500 |

**Secondary antibodies**

| Target species | Conjugate | Source | Dilution | Catalog # | Identifier |
|---|---|---|---|---|---|
| Anti-mouse | Cy5 | Jackson Immuno | 1:2000 | 715-175-151 | RRID:AB_2340820 |
| Anti-goat | Cy3 | Jackson Immuno | 1:2000 | 705-166-147 | RRID:AB_2340413 |
| Anti-chicken | Alexa 488 | Jackson Immuno | 1:2000 | 703-546-155 | RRID:AB_2340376 |
| Anti-chicken | Cy5 | Jackson Immuno | 1:2000 | 703-006-155 | RRID:AB_2340347 |

**Software and algorithms**

| Name | Source | Identifier |
|---|---|---|
| ZEN Blue/Black | Zeiss | RRID:SCR_013672 |
| Fiji | http://fiji.sc | RRID:SCR_002285 |
| Simple Neurite Tracer and SNT | https://imagej.net/SNT | RRID:SCR_016566 |
| Adobe Illustrator CS4 | Adobe | RRID:SCR_014198 |
| MATLAB | Mathworks | RRID:SCR_001622 |
| spyglass | IstoVisio | RRID:SCR_017961 |

with dental cement (C and B Metabond) for head stabilization. The head plate was then fixed in place by clamping head bars while a three-millimeter diameter circular region of the skull over the somatosensory cortex (–1.5 mm posterior and 3.5 mm lateral from bregma) was removed using a high-speed dental drill. A piece of the cover glass (VWR, No. 1) was placed in the craniotomy and sealed with cyanoacrylate glue (VetBond (3 M) and Krazy Glue).

**In vivo imaging**. Baseline images were acquired 2–3 weeks after cranial window installation, and mice with clear windows were randomly assigned to cuprizone or control conditions. Due to the rarity of animals having the correct genotype, successful intracranial viral injections, and clear windows, some animals with bone regrowth under the window at 2–3 weeks after the cranial window surgery had their windows repaired. Cyanoacrylate glue holding the original coverslip was carefully drilled away, the coverslip was removed, and invading bone into the window region was cut away before gluing in a new glass coverslip. Mice with repaired windows recovered for an additional week before checking the window clarity and taking baseline images. This procedure was performed for mice assigned to both control and cuprizone-treated groups.

During each imaging session (at baseline, 1 week recovery, and 5 weeks recovery), intracranially injected *PV-Cre; Ai9* mice were anesthetized with isoflurane (induction, 5%; maintenance, 1.5–2%, mixed with 0.5 L/min $O_2$ and fixed by their head plates in a custom stage. Two-photon images were collected using a Zeiss LSM 710 microscope equipped with a GaAsP detector using a mode-locked Ti:sapphire laser (Coherent Ultra) tuned to 1000 nm. The average power at the sample during imaging was <30 mW. Vascular landmarks and axon orientations were used to identify the same cortical area across imaging sessions. z stacks were 425 μm × 425 μm × 110 μm acquired at a resolution of 2048 × 2048 pixels using a coverslip-corrected Zeiss ×20 water-immersion objective (NA 1.0). Images of *Mobp-EGFP; PV-Cre; Ai9* mice were acquired as above, but z stacks were 425 μm × 425 μm × 230 μm.

Immediately after collecting the two-photon image, the sample was imaged again in reflectance mode to collect SCoRe signal with a z stack of the same dimensions as the two-photon image. The SCoRe has been shown by several groups independently to faithfully detect myelin sheaths in layer I of the cortex[9,10,52]. Laser parameters for reflectance imaging were used as previously described[9], and the pinhole was set to 1.5 Airy units.

Comparing SCoRe signal between baseline and recovery without visualization of oligodendrocyte cell bodies and processes meant surviving sheaths and sheaths lost and replaced in the same position were unable to be differentiated. While more frequent imaging may have allowed us to detect timepoints in which SCoRe signal was absent before reappearing in a subsequent time point, we found that weekly imaging of two-photon and SCoRe signal produced phototoxicity and increased the rate of axon damage. Our previous experiments using a 3-week cuprizone model found an average of 15.6% of all sheaths survive cuprizone in layer I[37]. Assuming these surviving sheaths would be randomly distributed across axon types, we estimate that surviving sheaths represent a very small fraction of the myelin analyzed and that the majority of overlapping SCoRe signal between baseline and five weeks recovery is remyelinated.

**Cuprizone administration**. At 9–11 weeks of age, male and female mice were fed a diet of milled, irradiated 18% protein rodent diet (Teklad Global) alone (control) or supplemented with 0.2% w/w bis(cyclohexanone) oxaldihydrazone (Cuprizone, Sigma-Aldrich) in gravity-fed food dispensers for three weeks. Both control and cuprizone-treated mice were returned to a regular pellet diet after three weeks during the recovery period[37].

**Image collection**. Images were acquired using a confocal laser-scanning microscope (Zeiss LSM 510 Meta; Zeiss LSM 710; Zeiss LSM 880). For population analyses (Fig. 2), 5 × 5 tiled z stacks (650 μm × 650 μm × 40 μm, z slice thickness: 0.5 μm, pinhole set to 1 Airy unit for each wavelength, 2048 × 2048 pixels) were acquired with a Zeiss ×63 oil objective (NA 1.4) in the primary somatosensory cortex in a cortical flatmount. Regional comparisons for PV axons used 2 × 2 tiled z stacks acquired at ×63. For diameter analysis, individual z stacks were acquired at ×63 in Zeiss Airyscan mode (59.65 μm × 59.65 μm or 78.01 μm × 78.01 μm, 2048 × 2048 pixels, z slice thickness: 0.21 μm, pinhole: 1 Airy unit). Tiled overviews of flatmounts and coronal sections were acquired at ×4 with a Keyence epifluorescence microscope.

**Diameter calculations**. Within high-resolution z stacks of each neuronal subtype, the distribution of gray values across the width of individual axons were averaged across a 2-μm length (ImageJ "Plot Profile"), and the full-width at half maximum (FWHM) was used to represent the diameter. Two to three of these measurements were taken for a 35-μm segment of axon within the imaging volume and averaged together for each individual axon. The probability myelinated for each axon was calculated as the number of measurement locations that were unmyelinated divided by the number of measurement locations that were myelinated. Given that the very small image area required for sufficient resolution (see the section "Image collection") was close to the average length of myelin internodes, the majority of axons had probability values of either 0 or 1.

**Axon and myelin tracing**. In all experiments, channels containing axon and myelin information were first split before tracing began to blind experimenters to the myelination status of each axon. Axons were selected for tracing by placing a 100 μm × 100 μm grid across the image and a random number generator selected grid coordinates for each axon seed. The first axon from the pia observed reaching a length of at least 100 μm (passed across the grid square) was selected and traced from that grid square along its full length within the image in both directions. Axon branches within the image were also traced if their lengths were also at least 100 μm in length. Axon traces were then imported into the myelin channel (either MBP immunolabeling or SCoRe signal) and used to trace myelin segments associated with each axon.

Fluorescently labeled axons were traced in Fiji using Simple Neurite Tracer/SNT[82]. Traced segments were put through a smoothing function prior to length calculations to reduce artifacts of jagged traces. As axons and their associated MBP segments were traced in their respective channels and signal properties differed between them, some continuously myelinated axons (completely covered by myelin

save for nodes of Ranvier) had PLM values > 100% if jitter in MBP traces was higher than the axon traces leading to an artefactually higher total sheath length than total axon length. In these cases, PLM was manually set to 99%.

*Mobp-EGFP* sheaths were traced in syGlass (v 1.6.0) virtual reality software using randomly selected 100 μm × 100 μm × 100 μm cubic ROIs (2 per ×20 imaged region) corresponding to ~130–230 μm depth from the pia. Subsequently, these traces were corrected and labeled as belonging to *PV-Cre; Ai9* or other axons in Fiji using SNT.

**Image processing and analysis**. Image stacks and time series were analyzed using Fiji. Images used in figures were adjusted for brightness and contrast levels for clarity. In vivo z stacks were de-noised with a 3D median filter (1-pixel radius). SCoRe images were first background subtracted (15-pixel rolling ball), 3D median filtered (1-pixel radius), and all three channels were summed together for a final SCoRe channel used for tracing. Because virally expressed EGFP signal from VM thalamocortical axons strongly bled through into the red channel at 1000 nm excitation, the green channel was subtracted from the red channel before tracing tdTomato-positive PV axons. Longitudinal image stacks were randomized for analysis and traced by a blinded experimenter and revised by a second blinded experimenter. Baseline and 5-week recovery time points were randomly assigned as "A" or "B" and experimenters always made original axon traces using the "A" time point. In some cases when the baseline was the "A" time point, traced axons were no longer visible in the recovery time point (either by part of the imaged region being obscured by meningeal thickening/bone growth or by axonal degeneration, which occurred at low rates in both control (VM: $8 \pm 5\%$, PV: $1 \pm 1\%$) and cuprizone-treated (VM:$14 \pm 3\%$, PV: $3 \pm 1\%$) animals). These rates of loss were not significantly different between control and cuprizone conditions (VM: $p = 0.38$, PV: $p = 0.46$, two-sample unpaired *t*-test). In these cases, missing axons were excluded from analysis. Axons that remained unmyelinated at both timepoints (PLM = 0) were not included in linear models so as not to strongly bias trends of myelin replacement.

**Statistical analysis**. Statistical analyses were performed with MATLAB (Mathworks) and Microsoft Excel. Significance was typically determined using Kruskal–Wallis one-way ANOVA with Dunn–Šidák correction for multiple comparisons. Violin plots were created using violin.m[83]. Each figure legend or the text otherwise contains the number of animals used, statistical tests used to measure significance, and the corresponding significance level (*p* value). Data are reported as mean ± standard error, or prediction [lower 95% confidence bound, upper 95% confidence bound], and $p < 0.05$ was considered statistically significant.

**Reporting summary**. Further information on research design is available in the Nature Research Reporting Summary linked to this article.

## Data availability
All published image data, code, tools, and reagents will be shared on an unrestricted basis; requests should be directed to the corresponding authors. Raw tracing data files are available at https://github.com/clcall/Call_Bergles_2021_CTSM[84] and summary data is included in the Source Data file. Source data are provided with this paper.

## Code availability
MATLAB scripts and ImageJ macros are available at https://github.com/clcall/Call_Bergles_2021_CTSM[84].

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

## Acknowledgements

We thank Dr. S.P. Brown and previous lab members for generously providing Cre lines used in this study, Dr. M. Pucak for technical assistance and impeccable instrument maintenance, T. Shelly for machining expertise, T. Bhardwaj and J.H. Kim for assistance in axon tracing, J. Orthmann-Murphy for assistance in cranial window surgeries, and members of the Bergles laboratory for insightful discussions. C. Call was supported by a National Science Foundation Graduate Research Fellowship (DGE-1746891). Funding was provided by grants from the NIH (NS051509, NS050274, NS080153), and the Dr. Miriam and Sheldon G. Adelson Medical Research Foundation to D. Bergles.

## Author contributions

C.L.C. conceived of and designed experiments, collected data, performed analyses, and wrote the manuscript. D.E.B. conceived of and designed experiments and wrote the manuscript.

## Competing interests

The authors declare no competing interests.
