## [Peer Review File · Nature Communications]

Reviewers' Comments:

Reviewer #1:

Remarks to the Author:

The authors have fully addressed my previous critiques with text changes and new analysis of data to help clarify points. The new title more appropriately summarizes the major points of the paper. I largely agreed with the other reviewers' initial comments, which I feel like the authors have done a sufficient job addressing. In conclusion, I am supportive of publication of this rigorously-conducted paper in *Nature Communications*.

Reviewer #2:

Remarks to the Author:

The revised manuscript by Call et al. is much improved, with changes to both the figures and the text adding clarity and strengthening their conclusions. This is a technically well-executed, highly rigorous, and thorough investigation into timely questions of neuronal subtype myelination and remyelination that would be an important addition to the field. My concerns have been addressed, and I have no additional suggestions.

Reviewer #3:

Remarks to the Author:

It remains unclear the extent to which remyelination reconstitutes the original pattern of myelin on axons and if more highly myelinated subtypes are more prone to remyelination. This manuscript from Call and Bergles uses a combination of neuronal subtype-specific labeling combined with myelin immunohistochemistry in Layer 1 to first describe the degree of myelination. Then they perform *in vivo* imaging to describe the remyelination pattern of select neuronal populations following cuprizone demyelination. They make three central findings:

1. Myelination patterns of different neuronal subtypes vary considerably within layer 1
2. The probability of myelination is a function of both axon caliber and neuronal subtype
3. Remyelination broadly reconstitutes the extent of myelin (albeit with different patterns) on two neuronal subtypes (PV and VM).

Although essentially a descriptive study, this paper is extremely well-written and the data support the authors' conclusions. These data provide a more granular description of the myelin content on axons from differing neuronal subtypes within the upper cortical layer and its association with axon diameter than previously existed. The final part of this paper attempts to address a critical question in remyelination biology – to what extent does remyelination reconstitute myelin content on specific neuronal populations? The paper provides a detailed description of the remyelination patterns of parvalbumin (PV+) and thalamocortical projections of the ventral medial (VM) nucleus. The authors have fully addressed my minor concerns from the previous submission (to *Nat Neuro.*) and I see this manuscript as being highly appropriate for publication in *Nat. Comms.* in its current form.